# RewardCode: Training Generalist Code Reward Model via Pairwise Reinforcement Learning

## Abstract

Test-time scaling improves code generation capacity of LLMs by leveraging a reward model to identify the best solution from multiple candidates. However, coding tasks span diverse domains, making unified evaluation challenging. In this paper, we present RewardCode, a generalist reward model for coding tasks. RewardCode performs principle-guided scoring, generates executable unit tests, and conducts pointwise evaluation of solutions, enabling scalable and fine-grained assessments. To train a cross-task code reward model, we construct CodePair-19K, a dataset of verifiable code preference pairs with task summaries and executable unit tests. Furthermore, we carefully design a two-stage training pipeline for RewardCode. The first stage combines Structural Summarize Fine-Tuning and Group Rejective Fine-Tuning, where diverse task descriptions are distilled into structured summaries to improve cross-domain code understanding and high-quality trajectories are bootstrapped through group rejection sampling from LLMs. The second stage introduces Pairwise-GRPO, a reinforcement learning method that leverages preference pairs to enhance the model's ability to distinguish between solutions while ensuring the generation of consistent and verifiable unit tests. Experiments on multiple benchmarks show that RewardCode outperforms strong baselines in accuracy and task success, proving its effectiveness in advancing general-purpose Code LLMs.

## 1 Introduction

Large Language Models (LLMs) Liu et al. (2024a); Achiam et al. (2023); Yang et al. (2025) have demonstrated remarkable performance across a wide range of coding tasks, including software engineering Jimenez et al. (2023), machine learning development Chan et al. (2024), and programming competitions Jain et al. (2024). Open-source code LLMs pretrained on large-scale code corpora, such as Qwen2.5-Coder Hui et al. (2024) and DeepSeek-Coder Guo et al. (2024), have become widely adopted in these domains. Recently, Test-Time Scaling (TTS) has emerged as a powerful approach to further enhance the performance of code LLMs by generating multiple candidate solutions and selecting the best one for execution.

To avoid the time-consuming process of executing candidate code, an accurate reward model is essential for efficiently identifying the best solution in TTS process. Generative Reward Models (GRMs) often rely on pairwise comparisons between candidate answers, repeatedly judging pairs to determine a final winner. However, this approach is computationally expensive and difficult to scale under large-scale TTS. In addition, coding tasks span heterogeneous domains, each requiring distinct domain expertise. This diversity makes unified evaluation highly challenging. Specialized code reward models, such as AceCodeRM Zeng et al., are capable of evaluating individual solutions. However, they are largely limited to relatively simple programming tasks and struggle to deliver accurate assessments in more complex domains, such as machine learning engineering. Motivated by these challenges, we raise the following question: ***Can we design a unified code reward model that works across diverse tasks?***

In this work, we propose RewardCode, a generalist code reward model that integrates principle-guided scoring with executable unit test generation. Complementary consensus filtering between

the two is used to derive the final evaluation results. Unlike pairwise reward models that rely on repeated comparisons, RewardCode evaluates candidate solutions in a pointwise manner, enabling scalable and fine-grained assessment across large collections of code. To support the training of a generalist code reward model, we construct CodePair-18K, a cross-domain dataset of verifiable preference pairs with structured task summaries and executable unit tests, collected from diverse domains including mathematical code translation, programming competitions, data science engineering, and software engineering. We design a two-stage training pipeline to train RewardCode. *In the first stage*, we propose Structural Summarize Fine-Tuning to transform diverse task descriptions into structured summaries for better cross-domain understanding, together with Group Rejective Fine-Tuning to bootstrap high-quality trajectories from large reasoning LLMs through group rejection sampling. *In the second stage*, a reinforcement learning method, Pairwise-GRPO, is designed, which leverages preference pairs with rule-based rewards to enhance the model's ability to discriminate between solutions and to generate consistent and verifiable unit tests.

We conduct extensive experiments to assess RewardCode across a variety of benchmarks. On large-scale code generation settings such as LiveCodeBench , RewardCode consistently improves Best-of-N accuracy and task completion rates over competitive baselines. We further evaluate on the code subset of RewardBench and RM-Bench, where RewardCode demonstrates stable performance as a unified code evaluator across benchmark. Overall, the results indicate that RewardCode can serve as a dependable component for test-time scaling in code generation and represents a step toward more general-purpose Code LLMs. Our main contributions are as follows:

- We introduce **RewardCode**, the first generalist code reward model that provides accurate and scalable evaluation from simple programming tasks to complex real-world scenarios such as Machine Learning and.

- We design **Pairwise-GRPO** to optimize scoring accuracy through relative score comparisons within preference pairs. It also prevents reward hacking by jointly validating unit tests, thereby enhancing the model's ability to discriminate between solutions and to generate consistent, verifiable tests.

- We conduct extensive experiments on mainstream and realistic benchmarks, including LiveCodeBench and and Reward Benchmarks, where RewardCode achieves superior Best-of-N accuracy and task success rates. We further demonstrate its effectiveness as a unified code evaluator on RewardBench and RM-Bench.

## 2 RELATED WORK

### 2.1 CODE LLM AND AGENT

Recent advances in large language models (LLMs) have led to significant progress in code generation, enabling the automation of a wide range of programming tasks. Foundation Code LLMs, pretrained on vast code corpora, have demonstrated strong performance across various domains such as software engineering, data science, and algorithmic programming Li et al. (2023); Guo et al. (2024); Hui et al. (2024). These models have fueled the development of benchmarks such as LiveCodeBench Jain et al. (2024) and BigCodeBench Zhuo et al. (2024), which are designed to systematically assess the capabilities of code generation models. Simultaneously, LLM-driven agents have been developed to tackle more complex real-world coding tasks in fields like software engineering and data science Yang et al. (2024); Zhang et al. (2025); Zeng et al. (2025); Wei et al. (2025); Liu et al. (2025a); Ou et al. (2025); Liu et al. (2025b). For instance, SWE-Agent Yang et al. (2024) introduces an agent-computer interface that significantly enhances capabilities for creating and editing files, navigating repositories, and executing tests. AIDE Jiang et al. (2025) enables rich exploration in the space of code by facilitating systematic search and adaptive refinement to improve task synthesis. ML-Master Liu et al. (2025b) integrates reasoning and exploration through a scoped memory mechanism, enabling effective automation of complex data science tasks. These agents require highly flexible and robust reward models to guide them in solving open-ended coding tasks.

## 2.2 REWARD MODEL

Reward models (RMs) have become essential for replacing human evaluators in assessing the quality of LLM outputs. Early reward models often produced scalar scores based on the output's quality, typically trained with the Bradley-Terry (BT) loss. However, these pointwise scores fail to capture the multi-dimensional aspects of code evaluation, such as correctness, efficiency, and test coverage. More recent developments in Generative Reward Models (GRMs) Mahan et al. (2024); Liu et al. (2025c); Chen et al. (2025c); Guo et al. (2025b); Chen et al. (2025a); Zhao et al. (2025b); Whitehouse et al. (2025) have explored integrating natural language critiques alongside evaluation signals. DeepSeek-GRM Liu et al. (2025c) introduces Self-Principled Critique Tuning (SPCT) to guide models in applying their own evaluation principles. Similarly, RM-R1 Chen et al. (2025c) frames reward modeling as a reasoning task, leveraging high-quality reasoning chains and verifiable rewards to improve evaluation quality. However, in code generation tasks, correctness is often measured by deterministic signals, such as unit tests, and incorporating natural language critiques may introduce inconsistencies or biases. To address this, CodeRM Ma et al. (2025) has explored the generation of unit tests via LLMs, but it remains limited to relatively simple code tasks. AceCodeRM Zeng et al. constructs code preference pairs to train scalar reward models, yet it is constrained to simpler programming challenges. In contrast, our work is the first to integrate GRMs into a unified evaluation framework for code. We propose a model that jointly supports scalar scoring, unit test generation, and critique generation, enabling accurate and generalizable reward modeling across diverse coding tasks.

## 2.3 REINFORCE LEARNING

Reinforcement learning (RL) has emerged as a key paradigm for improving LLMs by aligning them with specific goals or human preferences Schulman et al. (2017); Rafailov et al. (2023); Hu (2025); Yu et al. (2025). Early successes like RLHF demonstrated that human preference signals can effectively guide LLM behavior Ouyang et al. (2022). The latest progress, led by DeepSeek-R1 Guo et al. (2025a), introduced Reinforcement Learning from Verifiable Rewards (RLVR), which optimizes models using outcome-based signals, such as code execution results or unit test outcomes Zhao et al. (2025a); Wen et al. (2025). In code generation, RLVR typically relies on the results of interpreter execution or unit tests as verifiable rewards. In agent-based environments, Agent-RLVR Da et al. (2025) combines unit-test validation with pedagogical guidance and environment feedback to refine agent trajectories. DeepSWE Luo et al. (2025) relies purely on code execution outcomes and unit test results to train fully open-source coding agents. Beyond software engineering, ML-Agent Liu et al. (2025a) applies execution feedback and task success signals from machine learning workflows to reinforce autonomous ML engineering. These advancements demonstrate the potential of RL-based approaches to further align LLMs with real-world code generation tasks. However, the challenge remains to create reward models that can effectively evaluate complex and diverse coding tasks, especially when leveraging RL for test-time scaling and large-scale multi-task performance.

## 3 REWARDCODE

In this work, we develop RewardCode, a general code reward model designed to provide judgment across diverse programming tasks. Figure 1 presents the overall pipeline of RewardCode. We first construct CodePair-18K, a dataset of 18K validated code preference pairs with corresponding unit tests. RewardCode is trained in two stages. We begin by extracting task descriptions from CodePair-18K and employing a powerful LLM to generate structured summaries for each task. In addition, we collect high-quality trajectories from large reasoning models through group rule-based rejection sampling. These two sources are used to supervised fine-tune the base model, rapidly enhancing its ability to generate structured summaries and to perform general evaluation across different code tasks. Finally, we design Pairwise-GRPO to strengthen the model's capacity to autonomously discriminate between better and worse solutions when no mutual information is available, and to improve its ability to generate correct and consistent unit tests.

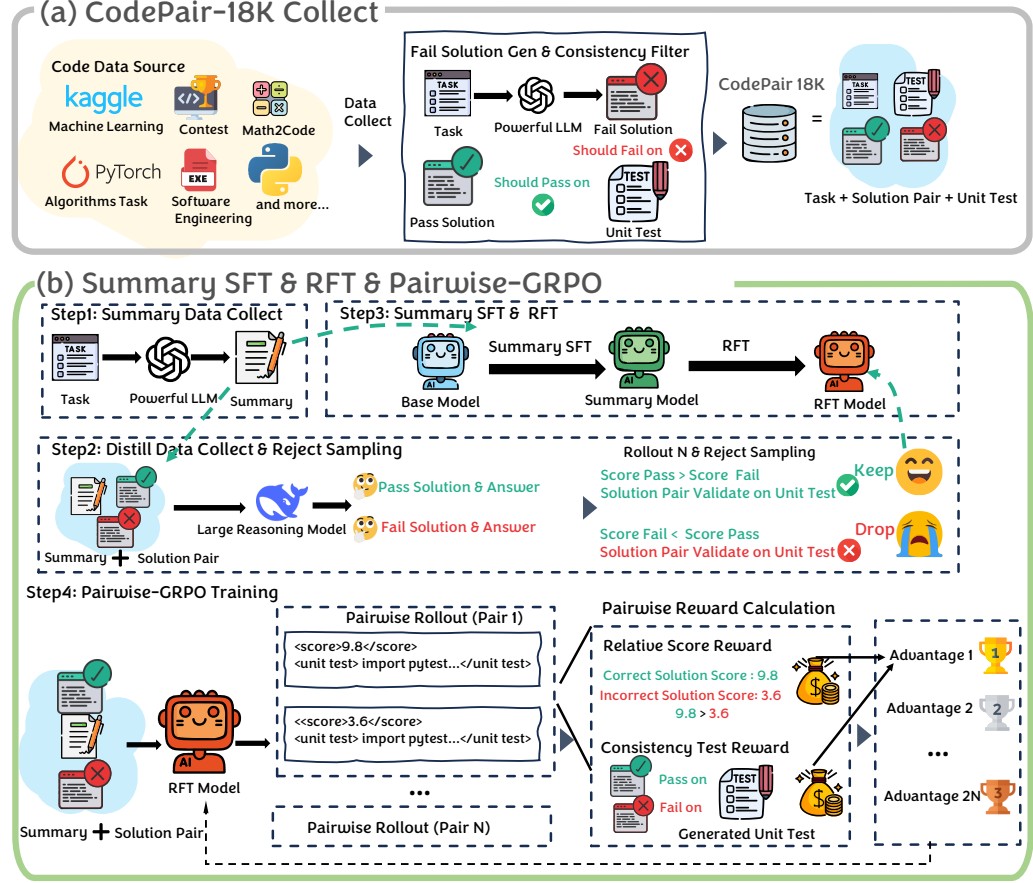

Figure 1: The overall pipeline of **REWARDCODE**. The two-stage training process begins with supervised fine-tuning a base model on the new CodePair-18K dataset and collected reasoning trajectories. In the second stage, the model is refined with Pairwise-GRPO to improve its ability to discriminate between code solutions and generate correct unit tests.

Table 1: Summary of existing code preference pair dataset.

| Datasets | Code Preference | Verification Code | Complex Task | Executable Unit Test | Correctness Verification | Function Design | Math2Code | Programming Contest | MLE | SWE | # Size |
|---|---|---|---|---|---|---|---|---|---|---|---|
| Code-Preference-Pairs | ✓ | ✗ | ✗ | ✗ | ✗ | ✓ | ✗ | ✗ | ✗ | ✗ | 54K |
| Target-DPO-59K | ✓ | ✓ | ✗ | ✗ | ✗ | ✓ | ✗ | ✗ | ✗ | ✗ | 59K |
| CodeDPO-114K | ✓ | ✓ | ✗ | ✗ | ✓ | ✓ | ✗ | ✗ | ✗ | ✗ | 114K |
| AceCodePair-300K | ✓ | ✓ | ✗ | ✗ | ✓ | ✓ | ✗ | ✗ | ✗ | ✗ | 300K |
| CodePair-18K | ✓ | ✓ | ✓ | ✓ | ✓ | ✓ | ✓ | ✓ | ✓ | ✓ | 18K |

## 3.1 CODEPAIR-18K DATA COLLECTION

To construct a cross-task general code reward model, we collect more than 18K code preference pairs with corresponding task descriptions. Specifically, we gather around 10K programming tasks covering a wide range of domains with increasing difficulty, including mathematical translation, algorithmic programming, programming contests, data science, and software engineering. Each task is paired with unit tests whenever available. For domains such as machine learning engineering , where executable unit tests are often missing, we employ Claude-4-Sonnet to generate quickly verifiable unit tests as ground-truth references. For tasks with verified correct solutions, we use GPT-4o to introduce subtle errors into correct code, thereby constructing the corresponding incorrect code and forming preference pairs. For tasks without gold-standard solutions, we generate multiple candidate implementations using LLMs and evaluate them with the ground-truth unit tests to assign correctness labels. Correct and incorrect solutions are then randomly paired to form preference pairs. Finally, we apply a Consistency Filter to ensure the reliability of each pair, retaining only

those where the correct code passes the unit tests and the incorrect code fails. Through this process, we obtain CodePair-18K, a cross-task and verifiable dataset of code preference pairs.

## 3.2 PRINCIPLE GUIDED SCORING AND TEST GENERATION

We formulate the general paradigm of code evaluation as follows: given a programming task and its candidate solution, a generalist code reward model should be able to generate both a reliable score and executable unit tests for verification. Based on this principle, we design RewardCode as a pointwise reward model, avoiding the limited applicability of pairwise RMs and the computational bottleneck in large-scale evaluation.

$$\{r, u\} = \pi_\theta(s, y), \tag{1}$$

where $s$ and $y$ represent the coding task and the corresponding solution, respectively, $r$ and $u$ represent the scoring result and the generated unit test. Unlike chat or math reasoning tasks, programming tasks allow verification through carefully designed unit tests, where the execution results directly determine correctness. Therefore, RewardCode adopts Principle Guided Scoring and Test Generation to produce both scores and tests. We design three intuitive principles for code evaluation: Task Completion, Code Correctness, and Efficiency. Task Completion checks whether the solution fulfills the required functionality, Code Correctness assesses the logical soundness of the code, and Efficiency evaluates the quality and resource efficiency of the implementation. RewardCode assigns individual scores for each principle and determines their weights according to the task category. The final score is the weighted sum of the three principle scores. Principle Guided Scoring actively encourages RewardCode to conduct detailed inspections of the code and strengthens their understanding of candidate solutions. After a thorough understanding, RewardCode generates directly executable unit tests according to the task objectives and code content, thereby validating correctness in practice. All unit tests are formalized in Pytest style, ensuring easy and executable verification. In this way, RewardCode can achieve efficient large-scale code evaluation through score comparison, while executable unit tests provide a reliable ground truth of correctness. During evaluation, we perform consensus filtering on the scores and unit tests. When making decisions on multiple targets, the candidate with the highest score among the tested candidates will be selected as the best answer.

## 3.3 STRUCTURAL SUMMARIZE AND REJECTIVE FINE-TUNING

To train RewardCode in Principle-Guided Scoring and Test Generation, we first collect a small cold-start dataset for supervised fine-tuning (SFT). This process consists of Structural Summarize Fine-Tuning and Rejective Fine-Tuning, which are designed to enhance the model's understanding of diverse coding tasks and its ability to generate reliable evaluations and tests.

**Structural Summarize Fine-Tuning.** Compared with other tasks, programming tasks contain explicit structural information such as inputs and outputs. However, task descriptions across different code domains vary significantly and are often task-specific. To enable the model to quickly understand heterogeneous programming tasks, we use GPT-4o to transform diverse task descriptions from CodePair-18K into structured summaries. Each summary includes the task objective, input–output specification, and other necessary details for solving the task. Compared with verbose natural-language descriptions, structured summaries provide RewardCode with concise and informative inputs, enabling it to quickly identify task goals and make more accurate judgments.

**Rejective Fine-Tuning.** The purpose of Rejective Fine-Tuning is to bootstrap RewardCode's ability to generate Principle-Guided Scoring and Unit Tests in the correct format by leveraging the reasoning capabilities of large models. Starting with CodePair-18K, we sample a subset of task summaries and their corresponding code solution pairs. For each task-solution pair, we use DeepSeek-R1-0528 to generate N responses, but without revealing whether the responses are correct or not. This allows us to simulate the model's reasoning capability without requiring explicit correctness labels. To filter out low-quality responses, we employ Group Rejective Sampling. For correct solutions, all generated responses must achieve higher scores than those associated with incorrect solutions. Conversely, for incorrect solutions, their responses must score lower than those of correct ones. This ensures that the model learns to distinguish between correct and incorrect solutions based on the underlying reasoning and output quality. In addition to scoring, we also generate unit

tests for each solution. The unit tests are validated by comparing their results with the expected outcomes, ensuring that the generated tests are both accurate and executable. Only unit tests that pass the validation step are retained, ensuring the quality of the test generation process. This step is crucial for teaching RewardCode to generate reliable unit tests that align with its scoring mechanism, reinforcing the model's ability to evaluate code accurately and comprehensively.

### 3.4 PAIRWISE-GRPO

We propose Pairwise-GRPO to leverage large-scale code preference pairs and further enhance RewardCode's ability in Principle Guided Scoring and Test Generation. Specifically, for the input code pair dataset $D = (I_i, y_a, y_b)_{i=1}^N$, RewardCode generates scores and unit tests for each solution in a pointwise manner:

$$(\{r_a, u_a\}, \{r_b, u_b\}) = (\pi_\theta(s_i, y_a), \pi_\theta(s_i, y_b)), \tag{2}$$

where $s_i, y_a, y_b$ denote task summary, correct solution and incorrect solution. $\pi_\theta$ represents RewardCode model. $\{r_a, u_a\}, \{r_b, u_b\}$ correspond to the score and unit test generated for the correct and incorrect solutions, respectively. We then compute rewards in a pairwise fashion. First, we define the Relative Score Reward $R_{score}$, which evaluates whether the relative ranking of $r_a$ and $r_b$ aligns with ground-truth correctness:

$$R_{score} = \begin{cases} 1 & \text{if } r_a > r_b, \\ 0 & \text{otherwise.} \end{cases} \tag{3}$$

The Relative Score Reward accurately reflects whether RewardCode can provide correct preference judgments in the absence of explicit relative information, thereby encouraging the model to produce more precise scores. To validate the quality of unit tests, we design the Unit Test Reward, which encourages the model to generate tests that pass on the correct solution and fail on the incorrect one. The Unit Test Reward of $u_a$ is represented as:

$$R_{ut}^a = \begin{cases} 0.5, & \text{if } P(r_a, u_a) = 1 \text{ and } P(r_b, u_a) = 0, \\ 0, & \text{otherwise,} \end{cases} \tag{4}$$

where $P$ denotes the execution result of a unit test. The Group Unit Test Reward is given by $R_{ut} = R_{ut}^a + R_{ut}^b$. The final pairwise reward $R$ is calculated as:

$$R = R_{score} + R_{ut}^a + R_{ut}^b + R_{format}, \tag{5}$$

where $R_{format}$ is structural format reward. After collecting rollout samples for each code pair and their corresponding rewards, we optimize the policy iteratively with the following objective:

$$
\begin{aligned}
J(\theta, \{o_i\}_{i=1}^G) = \mathbb{E}_{\substack{q \sim P(Q) \\ \{o_i\} \sim \pi_{\theta_{\text{old}}}(\cdot|q)}} & \left[ \frac{1}{G} \sum_{i=1}^G \min\left[ \frac{\pi_\theta(o_i \mid q)}{\pi_{\theta_{\text{old}}}(o_i \mid q)} A_{o_i}, \text{clip}\left( \frac{\pi_\theta(o_i \mid q)}{\pi_{\theta_{\text{old}}}(o_i \mid q)}, \varepsilon \right) A_{o_i} \right] \right] \\
& - \mathbb{E}_{\substack{q \sim P(Q) \\ \{o_i\} \sim \pi_{\theta_{\text{old}}}(\cdot|q)}} \left[ \beta \, \mathrm{D}_{\text{KL}}[\pi_\theta \| \pi_{\text{ref}}] \right],
\end{aligned}
\tag{6}
$$

where $\text{clip}(x, \varepsilon) := \min(\max(x, 1 - \varepsilon), 1 + \varepsilon)$, $\pi_\theta$ is the policy to be optimized, $\pi_{\text{old}}$ is the old policy, $\{o_i\}_{i=1}^G$ are the pairwise rollout, and $A_{o_i}$ is the normalized pairwise reward related to $R_{o_i}$.

## 4 EXPERIMENT

### 4.1 EXPERIMENTAL SETTINGS

#### 4.1.1 EVALUATION METRIC AND BENCHMARK

We evaluate RewardCode across diverse programming scenarios. Experiments are conducted on mainstream code programming benchmarks, including LiveCodeBench Jain et al. (2024). For LLM-based benchmarks, we adopt Qwen2.5-Coder-7B-Instruct as the code LLM to generate 16 candidate solutions, from which the reward model selects the best solution. Our primary metric is The successful rate of answers after Best-of-N selection, which measures a reward model's ability to identify correct solutions under test-time scaling. We further evaluate RewardCode on reward benchmarks with code-specific subsets, including RewardBench Lambert et al. (2024), RM-BenchLiu et al. (2024b), and RMBZhou et al. (2024). These reward model benchmarks contain pre-collected preference responses, which can be used to quickly assess the ability of different reward models.

### 4.1.2 MODEL IMPLEMENTATION AND OPTIMIZATION

We use Qwen2.5-Coder-7B-Instruct Hui et al. (2024) as the base model for RewardCode. For Pairwise-GRPO optimization, we set the learning rate to $1 \times 10^{-6}$ and and the KL coefficient to 0.001. RL Training is performed for 2 epochs over the full CodePair-18K dataset.

### 4.1.3 BASELINES

For BoN accuracy evaluation on code generation benchmarks, we compare RewardCode with four categories of reward models reflecting different design paradigms: For BoN accuracy evaluation on code benchmarks, we compare RewardCode with four categories of reward models reflecting different design paradigms, including (1)Scalar Reward Model: Internlm2-7b-reward, Eurus-RM-7b, RM-Mistral-7B (2) Generative Reward Model: JudgeLRM Chen et al. (2025b), RRM-7B Guo et al. (2025b), RM-R1-Qwen2.5-Instruct-7B Chen et al. (2025c). (3) Code Reward Model: AceCodeRM-7B Zeng et al..

## 4.2 RESULTS AND ANALYSIS

### 4.2.1 BEST-OF-N EVALUATION

| Model | RewardBench-Code | RM-Bench-Code | LiveCodeBench |
|---|---|---|---|
| *Scalar Reward Models* | | | |
| Eurus-RM-7b | 0.600 | 0.496 | 0.369 |
| RM-Mistral-7B | 0.940 | 0.530 | 0.304 |
| Internlm2-7b-reward | 0.941 | 0.502 | 0.3436 |
| *Generative Reward Models* | | | |
| JudgeLRM-7B | 0.820 | 0.352 | 0.363 |
| RM-R1-Qwen2.5-Inst-7B | 0.888 | 0.568 | 0.362 |
| RRM-7B | 0.849 | 0.528 | 0.407 |
| *Code Reward Models* | | | |
| AceCoderRM-7B | 0.520 | 0.613 | 0.404 |
| *Our Method* | | | |
| **RewardCode-7B** | **0.949** | **0.621** | **0.418** |

Table 2: Judge Evaluation Results on LLM-based code Benchmarks and the code subset of Reward Model Benchmarks.

**Judge Evaluation on LLM-based code Benchmarks.** We begin by comparing the ability of different reward models to select the best solution in Best-of-N (BoN) evaluations on code benchmarks, focusing on LiveCodeBench. The greedy result represents the accuracy of the policy model when the temperature is set to 0, generating a single sample to reflect the model's performance on the corresponding coding task. We find that RewardCode outperforms all baseline methods overall. Scalar RMs struggle with real code evaluation, often performing worse than greedy results, which suggests that scalar models are not suitable for real-world code evaluation. While GRMs perform better than scalar RMs, their improvement is limited, and their reliance on pairwise input comparisons becomes computationally expensive as the number of TTS candidates increases. In contrast, specialized code RMs trained on code preferences perform better, though their improvement in TTS performance remains limited on more diverse code distributions, suggesting that existing code reward models still have room for improvement in multi-scenario tasks. Compared to these baselines, RewardCode consistently delivers the best performance across multiple benchmarks, demonstrating its ability to benefit from diverse code RL training. We also evaluate each RM on a code subset of RewardBench and RM-Bench to reflect RM's judgment capabilities on code. While scalar RMs perform better on these RM Benchmarks than in real code BoN evaluations, generative RMs lag behind, likely due to scalar RMs being trained on a larger scale of relevant preference data. Notably, the code subsets in RewardBench and RM-Bench come from various programming languages, while RewardCode and other code RMs are mainly trained on Python data. Therefore, these evaluations test Code RM's

out-of-domain capabilities. As shown in Table 2, RewardCode-7B consistently outperforms other Code Reward Models across different RM Benchmarks, demonstrating its excellent generalization ability and the potential to transfer evaluation capabilities trained on Python to multiple languages.

**Best-of-N on Agent-based code Benchmarks.**

### 4.2.2 ABLATION STUDY

| Model | RM-Bench-Code | LiveCodeBench |
|---|---|---|
| *RewardCode-Variant* | | |
| RewardCode-Zero | 0.589 | 0.366 |
| RewardCode-SFT | 0.582 | 0.387 |
| RewardCode-Score | 0.603 | 0.405 |
| RewardCode-UT | 0.591 | 0.347 |
| *Our Method* | | |
| RewardCode-7B | **0.621** | **0.418** |

Table 3: Ablation study of different RewardCode variants.

We conduct ablation experiments on BigCodeBench and LiveCodeBench to investigate the effectiveness of each design principle of RewardCode. The experimental setup follows that of Section 4.1. The ablation study includes four variants of RewardCode: (1) RewardCode-Zero, (2) RewardCode-SFT, (3) RewardCode-Score, and (4) RewardCode-UT. RewardCode-Zero represents the model optimized using only Pairwise-GRPO without Structural Summarize Fine-Tuning and Rejective Fine-Tuning. RewardCode-SFT is the version after applying Structural Summarize Fine-Tuning and Rejective Fine-Tuning. RewardCode-Score and RewardCode-UT represent models using only score-based and unit test-based judgments, respectively. From Table 3, we can see that RewardCode outperforms all variants. RewardCode-Zero performs the worst on Livecodebench because it has not undergone a cold start stage and the model itself lacks the ability to solve complex tasks. However, the SFT phase allows RewardCode to distill additional capabilities from powerful reasoning models, such as more accurate scoring and better unit test generation. Thus, the absence of the SFT phase limits the performance potential of RewardCode-Zero. RewardCode-SFT performs worse, indicating that without Pairwise-GRPO training, RewardCode-SFT cannot fully activate the distillation capabilities. This further emphasizes the significance of Pairwise-GRPO in guiding the model to generate higher-quality evaluations. Lastly, both RewardCode-Score and RewardCode-UT show some performance decline, suggesting that combining both score and unit test as a unified code evaluation paradigm is reasonable. This consensus filtering helps RewardCode avoid missing details with score-only evaluations and false positives with unit test-only evaluations, thus maximizing the performance of policy models in TTS.

## 5 CONCLUSION

In this paper, we introduce RewardCode, a generalist code reward model designed to provide precise and scalable code evaluations across diverse programming tasks. RewardCode adopts principle-guided scoring and test generation, with complementary consensus filtering used to derive the final evaluation results. We design Pairwise-GRPO to jointly optimize score and unit test generation, which obtains a pairwise shared reward to ensure the correctness of the scoring process and prevent false positives in unit tests generation. We also present CodePair-18K, a cross-domain dataset of verifiable code preference pairs with task summaries and executable unit tests, which forms the foundation for training RewardCode. Through extensive experiments on multiple benchmarks, RewardCode consistently outperforms strong baselines, achieving superior Best-of-N accuracy and task success rates. These results highlight the effectiveness of RewardCode as a generalist evaluator for code tasks and demonstrate its potential for future applications in more complex code domains.

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

# A APPENDIX

## A.1 USE OF LLMS

The LLMs were used as an assistive tool in the preparation of this manuscript. Its primary functions were writing polishing, including refining grammar and improving sentence structure for a consistent academic tone, and LaTeX code modifications, which involved generating and debugging code for the presented algorithms and tables. All content and code generated or modified by the LLM were thoroughly reviewed and verified by the authors to ensure scientific accuracy and correctness.

