# OpenReview forum: "RewardCode: Training Generalist Code Reward Model via Pairwise Reinforcement Learning"
_ICLR.cc/2026/Conference — ICLR 2026 Conference Withdrawn Submission_

### Official Review · Reviewer_9xxM · 2025-10-30

**Soundness:** 2
**Presentation:** 2
**Contribution:** 2
**Rating:** 4
**Confidence:** 3

**Summary:**

RewardCode is a general-purpose judge for code. It scores each candidate solution on its own and can even write runnable PyTest unit tests to check it. It’s trained in two phases: first with supervised “structural summaries” plus rejective fine-tuning, then with a new Pairwise-GRPO RL step that teaches it to rank correct answers higher and produce good tests. On LiveCodeBench and the code portions of RewardBench and RM-Bench, it beats strong scalar, generative, and code-specific reward baselines for Best-of-N selection.

**Strengths:**

Overall, the paper’s strengths are its practicality and clear evidence. RewardCode scores each candidate on its own—so it scales well to large Best-of-N runs—and it backs those scores with runnable PyTest checks. The training recipe (structured-summary SFT plus Pairwise-GRPO) is thoughtful, supported by an execution-verified, multi-domain dataset. Results are strong on LiveCodeBench and the code slices of RewardBench/RM-Bench, with ablations showing both the scoring and test-generation pieces really matter—and it even generalizes beyond Python.

**Weaknesses:**

- Heavy synthetic supervision. Many “ground-truth” labels come from LLMs (generated unit tests; perturbed incorrect solutions), so coverage and bias in real-world scenarios are open questions.
- Smallish dataset. CodePair-18K is careful but much smaller than some rivals, which may limit long-tail robustness.
- Benchmark coupling. Results are shown for Best-of-N using one generator and a fixed N=16; robustness to other policy models, temperatures, or larger N isn’t demonstrated.
- Modest absolute gains on LiveCodeBench. The win over strong baselines is narrow, so practical impact may hinge on cost/latency trade-offs that aren’t reported.
- Language skew. Training leans heavily on Python while benchmarks span multiple languages; cross-language generalization isn’t deeply stress-tested.
- Missing human evaluation and test-coverage analysis. The paper relies on automated benchmarks and doesn’t report human studies or quantitative coverage metrics for the generated tests.

**Questions:**

- Missing space after period: “REWARDCODE.The” in the Figure 1 caption.
- Duplicate sentence in Baselines: the line starting “For BoN accuracy evaluation…” is repeated.
- Model name inconsistency: “AceCodeRM-7B” vs. “AceCoderRM-7B” (choose one).

others see weakness.

---

### Official Review · Reviewer_JXir · 2025-10-31

**Soundness:** 2
**Presentation:** 3
**Contribution:** 2
**Rating:** 4
**Confidence:** 4

**Summary:**

This paper proposes RewardCode, a generalist reward model (RM) for coding tasks designed to improve Test-Time Scaling by selecting the best solution from multiple candidates. The model is designed to perform pointwise evaluation, provide principle-guided scores, and generate executable unit tests. To train this model, the authors created a new dataset, CodePair-18K, containing code preference pairs with unit tests. The training involves a two-stage pipeline: (1) SFT using Structural Summarize Fine-Tuning and Group Rejective Fine-Tuning to bootstrap evaluation capabilities; and (2) RL using a novel method called Pairwise-GRPO, which uses a combined reward signal based on relative scores and unit test consistency to optimize the model. Experiments on benchmarks like LiveCodeBench, RewardBench, and RM-Bench show that RewardCode outperforms baseline RMs in Best-of-N selection and preference accuracy.

**Strengths:**

1. Provide a reward model for diverse coding tasks.
2. The ablation study, while flawed, disentangles the scoring and unit-test generation components.

**Weaknesses:**

This paper suffers from several weaknesses in its methodology, presentation, and evaluation:

1. CodePair-18K relies on an unsound construction method. For complex domains like machine learning, where ground-truth unit tests are missing, the authors used Claude-4-Sonnet, a proprietary LLM, to generate the "ground-truth references". This means the training data for the novel unit-test generation task is not human-verified ground truth, but rather the output of a different black-box model. This dependency severely undermines the validity of the dataset and any model trained on it.

2. The primary experimental results do not support the paper's strong claims of superiority. In Table 2, the full RewardCode-7B model achieves a 0.418 on LiveCodeBench. This is a negligible improvement over the strongest baselines, RRM-7B (0.407) and the specialized AceCoderRM-7B (0.404). A ~1% improvement in accuracy does not justify the paper's complex two-stage training pipeline and claims of advancing general-purpose Code LLMs.

3. The ablation study in Table 3  critically undermines the paper's central thesis. The core contribution is the joint generation of scores and unit tests, combined via "consensus filtering". However, the RewardCode-Score variant (which removes the unit test generation entirely) achieves a score of 0.405. This is almost identical to the full model's 0.418 and on par with the baseline AceCoderRM (0.404). This demonstrates that the paper's main novel component, the generation and filtering of unit tests, provides almost no practical benefit.

4. The proposed training pipeline repackages existing techniques. Stage 1's "Structural Summarize Fine-Tuning" and "Group Rejective Fine-Tuning" are standard practices of distilling knowledge from stronger teacher models (GPT-4o and DeepSeek-R1). Stage 2's "Pairwise-GRPO" is an RL algorithm using a straightforward reward function composed of a standard preference-ranking objective ($R_{score}$) and a simple outcome-based reward for unit test execution ($R_{ut}$). The paper fails to demonstrate significant novelty beyond combining these standard components.

Minor:
line 68: such as LiveCodeBench , -> such as LiveCodeBench,
CodePair-19K or CodePair-18K

**Questions:**

See above

---

### Official Review · Reviewer_EAy8 · 2025-10-31

**Soundness:** 2
**Presentation:** 3
**Contribution:** 2
**Rating:** 4
**Confidence:** 4

**Summary:**

This paper proposes RewardCode, a generalist reward model for code evaluation across diverse programming tasks. The key technical contributions are: (1) A pointwise reward model that generates both principle-guided scores and executable unit tests, (2) Pairwise-GRPO, a novel reinforcement learning method that leverages preference pairs with rule-based rewards, and (3) CodePair-18K, a dataset of 18K verifiable code preference pairs spanning multiple domains. The model is trained in two stages: first with Structural Summarize Fine-Tuning and Group Rejective Fine-Tuning, then with Pairwise-GRPO to enhance discrimination ability.

**Strengths:**

- The Pairwise-GRPO method represents a genuinely novel combination of GRPO with pairwise preference learning for reward model training.
- The experimental results demonstrate strong performance, achieving best scores on LiveCodeBench and RewardBench-Code, outperforming both scalar and generative reward models.

**Weaknesses:**

- The paper doesn't justify why principle-guided scoring is needed when executable test cases already provide ground truth. For LiveCodeBench-style benchmarks, test execution is definitive.
- The 0.418 vs 0.404 improvement over AceCodeRM-7B doesn't justify the added complexity of the two-stage training and Pairwise-GRPO.
- Without showing how RewardCode improves actual policy models when used for RL stage, we can't verify if the reward model actually provides useful training signals.

**Questions:**

- Have you considered using this reward model to train target models during the RL phase to further validate the effectiveness of your approach? This would demonstrate whether RewardCode provides useful training signals for policy improvement.
- Could you add a control experiment where only test case pass rates are used as rewards? This comparison would clarify the added value of principle-guided scoring and unit test generation beyond simple execution-based verification.

---

### Official Review · Reviewer_r4mR · 2025-11-01

**Soundness:** 2
**Presentation:** 3
**Contribution:** 3
**Rating:** 4
**Confidence:** 4

**Summary:**

This paper introduces a reward model, RewardCode, that is designed to generalize over various coding domains unlike weaker scalar reward models. The first main contribution is CodePair-18K Collect, a preference dataset created by gathering code from diverse domains and data augmentations via LLMs. The second main contribution is RewardCode’s training pipeline, which involves (1) SFT to turn task descriptions into a model-generated summary and RFT to train on correct solutions and validated unit tests and (2) pair-wise GRPO atop the scores generated by Principled Guide Scoring and generated unit tests.

**Strengths:**

$Originality$: This paper introduces Principle Guided Scoring and Test Generation which requires the reward model to simultaneously output a score and test case which is novel.

$Quality$: The diversity of the dataset contribution could be helpful for improving reward model’s generalizability across domains.

$Clarity$: The motivation of the paper, construction of the dataset and experiments section are fairly clear.

$Significance$: The main weakness of reward models is their failure to generalize across domains; this paper is addressing a crucial problem.

**Weaknesses:**

- The figure was fairly confusing at the beginning, especially with the green lines pointing from Step 2 to Step 3 implying the inputs of the SFT/RFT stages.
- There is some missing information in the problem formulation, particularly how the Task Completion, Code Correctness, and Efficiency scores are computed
- The experiments section does not include sufficient evidence that RewardCode’s design makes it generalizable across code domains

**Questions:**

My main concern is regarding whether RewardCode can truly generalize across code domains. The results are missing experiments that support this claim.
- RewardCode-7B performs similarly to AceCoderRM-7B except on RewardBench-Code where there is a significant increase in performance; however, Table 1 shows that CodePair-18K is more diverse than AceCodePair-300K. Is this the cause of the improvement on RewardCode-7B?
   - How were the code data sources for CodePair-18K selected?
- The ablation is missing baselines and contradicts itself
  - How does the base model, Qwen2.5-Coder-7B-Instruct, perform as a Scalar Reward Model using CodePair-18K as the preference dataset? As a generative reward model?
  - “RewardCode-Zero performs the worst on LiveCodeBench because it has not undergone a cold start stage and the model itself lacks the ability to solve complex”
    - RewardCode-UT performs the worst on LiveCodeBench despite going through SFT.
  - “RewardCode-SFT is the version after applying Structural Summarize Fine-Tuning and Rejective Fine-Tuning”
    - If RewardCode-Zero is only optimized using Pairwise-GRPO, then why is RewardCode-SFT and RewardCode-7B not the same model?
  - How does RewardCode perform by ablating away the Structural Summarize Fine-Tuning only? Ablating away only the RFT?

In summary, I am still unclear on whether (1) the diversity of the dataset leads to an improvement over AceCoderRM-7B (2) what 1-1 baselines to compare RewardCode to given the same training dataset and (3) how the training stages and new training paradigms are contributing to the improvements in performance.

---

### Note · Authors · 2025-12-10

I have read and agree with the venue's withdrawal policy on behalf of myself and my co-authors.